# Decoding Stumpers: Large Language Models vs. Human Problem-Solvers

**Alon Goldstein**[1]  **Miriam Havin**[2]  **Roi Reichart**[3]  **Ariel Goldstein**[1245]

**\* Corresponding Author: alon@xoltar.com**

1 Xoltar Inc
2 Department of Cognitive and Brain Sciences, Hebrew University, Jerusalem, Israel
3 Faculty of Data and Decision Sciences, Technion
4 The Hebrew University Business School, Jerusalem, Israel
5 Google Research

## Abstract

This paper investigates the problem-solving capabilities of Large Language Models (LLMs) by evaluating their performance on stumpers, unique single-step intuition problems that pose challenges for human solvers but are easily verifiable. We compare the performance of four state-of-the-art LLMs (Davinci-2, Davinci-3, GPT-3.5-Turbo, GPT-4) to human participants. Our findings reveal that the new-generation LLMs excel in solving stumpers and surpass human performance. However, humans exhibit superior skills in verifying solutions to the same problems. This research enhances our understanding of LLMs' cognitive abilities and provides insights for enhancing their problem-solving potential across various domains[1]

## 1 Introduction

Since their inception, Large Language Models (LLMs) have astonished the scientific community with their ability to tackle complex tasks (Radford et al., 2019; Brown et al., 2020; Devlin et al., 2018). These emerging capabilities, along with shared principles with human cognition and the brain, have motivated significant efforts to utilize deep language models and, recently, LLMs for explaining neural activity (Tikochinski et al., 2023; Goldstein et al., 2022b,a; Schwartz et al., 2019), predicting human behavior (Goldstein et al., 2022b; Brand et al., 2023), and even providing a theoretical framework for the human mind (Richards et al., 2019; Hasson et al., 2020). Recent advancements, particularly the ability of LLMs to perform tasks requiring different skills such as mathematical calculations, analytical reasoning and use of world knowledge, have led several papers to declare that LLMs possess what is termed in the cognitive literature *System 2* capabilities (Matsuo, 2020; Kojima et al., 2022). The dual-system model of the mind has arguably been the most prevalent model of thought and behavior in psychology, economics and social science in general (Goldstein and Young, 2022; Evans and Stanovich, 2013; Chaiken and Trope, 1999; Gawronski and Creighton, 2013), especially in addressing systematic limitations of cognitive and artificial systems. In simple terms, *System 1* is associated with effortless, associative processes and is often thought of as compatible with neural nets implementation, while System 2 is related to effortful, serial, and often symbolic processes (Frankish, 2010; Evans, 2003). A famous example where System 1's heuristic hinders a solution is described in Box 1.

> **Box 1: The bat and the ball**
>
> "A bat and a ball cost 1.10 dollars in total. The bat costs 1 dollar more than the ball. How much does the ball cost?"
> The immediate but incorrect response is to assume that the ball costs 10 cents. However, a symbolic-serial approach formulation of the problem "$x + (x + 1) = 1.10$" yields the correct solution of 0.05 dollars.

While this type of questions (Cognitive Reflective Test; CRT) are considered hard to solve, as they tend to elicit wrong responses (Frederick, 2005; Toplak et al., 2011), people can be primed to solve them correctly by insisting on a formalist approach (i.e., applying System 2 instead of System 1; (Alter et al., 2007)). In contrast, problems that require insight (i.e., have neither an intuitive/associative solution nor a symbolic one) are hard for humans and often elicit no response (i.e., humans are "stuck"; (Bar-Hillel et al., 2018; Bar-Hillel, 2021)).

---

[1]The data is available at https://github.com/Alon-Go/Stumpers-LLMs.

Consider, for example, the riddle in Box 2:

> **Box 2: Blood relatives**
>
> "Alex is Bobbie's blood relative, and Bobbie is Charlie's blood relative, but Alex is not a blood relative of Charlie. How come?".
> Answer: Alex and Charlie can be related to Bobbie from different sides of the family, e.g., they could be his parents, uncles, etc.

This riddle typically challenges human intuition (System 1) because humans tend to consider Alex, Bobbie, and Charlie as blood relatives. However, a symbolic solution (System 2) is typically also not available to humans who try to solve it, as there is no clear algorithm to follow to reach a solution. Facing this question, humans seem to be anchored (or stuck) in the framing according to which the three men are blood relatives and cannot escape it to generate an alternative framing of the problem that would yield effective explanations of the situation (Bar-Hillel, 2021).

The above-mentioned question is an example of a *stumper*. A stumper is a one-step intuition riddle, the solution to which is typically so elusive that it does not come to mind, at least initially - leaving the responder stumped. Stumpers do not fall within the System 1 or System 2 frameworks but are related to creative thinking (Bar-Hillel et al., 2019). Importantly, once presented with a solution, people can easily classify it as right or wrong - a simple system-2 task. In this paper, we demonstrate that recent LLMs (e.g., GPT-3.5 and GPT-4) outperform humans in solving stumpers but lag behind humans in classifying solutions as right or wrong.

## 2   Task

A stumper is a single-step intuition riddle in which a misleading cue sparks a visual or semantic representation that blocks the solution from coming to mind. Unlike other riddles, there is no need for further computation or thinking, and once the obstacle is removed, the answer is clear. See examples in *Appendix A*.

The dataset used for our analysis consists of all 76 stumpers curated in (Bar-Hillel, 2021). Each stumper is a textual description of a narrative or scenario that requires a unique solution. To enhance the number of stumpers beyond this exhaustive list, we generated two similar riddles for each

stumper, by asking GPT-3.5-Turbo to change the names and wording of the original data-set. After the generation, we manually approved or edited each stumper to reduce confusion and alternative solutions as much as we could. This process resulted in a set of additional 152 stumpers. As the set of new stumpers was not validated with human participants and may differ from the original set, we present the results for the original set in the body of this paper and the detailed results in the appendix. The data also includes correct and incorrect solutions, which allowed a comparative analysis of the accuracy and reasoning strategies of the responses given by models and human participants. A dataset sample can be found in *Appendix A*.

## 3   Models and Experiments

The study involved four language models: Davinci-2, Davinci-3, GPT-3.5-turbo, and GPT-4. Additionally, 81 human participants (F=48%; ages 20-54, m=28.52, sd=7.73) were recruited via an online survey participation platform (Prolific).

When solving each stumper, both humans and models were presented with a prompt. To normalize the answers across conditions, models, and participants, all prompts started with a standardized definition of a correct answer to a riddle:

> An answer to a riddle is correct only if it is consistent with all the riddle's clues, sensical, specific, logical, and fitting with the context of the riddle.

See prompts examples in *Appendix C*.

### 3.1   Answer Generation

To avoid learning, each participant was presented with only one stumper and was asked to answer it or type "IDK" if they did not know the answer. Each model, in each prompt, was presented either with only one stumper ("naïve response") or with two other pairs of riddles and their ground-truth answer ("prompted response").

### 3.2   Answer Verification

After answering the riddle or typing "IDK", human participants were presented with the two possible solutions and were asked to choose the correct one. The models were presented with the same choice without their previous response in the prompt.

### 3.3 Answer Verification - Models' response

To further compare, the models were given a verification problem where their own answers replaced one of the answers. For riddles to which the model knew (/did not know) the answer, their response was used instead of the correct (/incorrect) ground truth.

## 4 Results and Observations

We replaced participants who reported knowing their riddle or finding the answer online. Two authors evaluated the responses unanimously, with only one response being disagreed upon, leading to its exclusion.

**LLMs proficiency at solving stumpers** Our results are provided in *Figure 1*. Solving a stumper by chance has virtually zero probability, given the infinitesimal likelihood of randomly arriving at the correct solution among countless potential answers. Human participants in our sample have accurately solved 38.15% of the stumpers, replicating the 35% accuracy found in (Bar-Hillel, 2021).

**Improved performance of the advanced models** A two-way ANOVA was conducted to examine the effects of the models and the prompting: The chat models (GPT-3.5-Turbo and GPT-4; m=57.8% for the original stumper, 43.4% for the enhanced data-set) have performed significantly better than the GPT-3 model (Davinci-2 and Davinci-3; m=29.6% for the original stumper, 22.5% for the enhanced data-set) [F(1,4)=686, p=0.00001]. The main effect of Prompt was not significant [F(1, 4) = 0.023, p = 0.887], but the interaction was [F(1, 4) = 30.82, p = 0.005], indicating that the prompt has a positive impact on the performance of the GPT-3 models and a negative impact on the performance of the chat models. See results for the original data-set in *figure 1*. See results for the enhanced set in *Appendix B*.

**Answer Verification** The answer verification task was tested once with the ground-truth answers (Figure 2) and once with the model's responses vs. ground truth (Figure 3). Different scores are reported for correct and incorrect responses.

While humans performed perfectly at verification when knowing to generate the answer (100%) and above chance even when they failed to generate (63.8%), most models performed below the chance level (m=41%). A two-way ANOVA was conducted to compare the models' ability to choose the

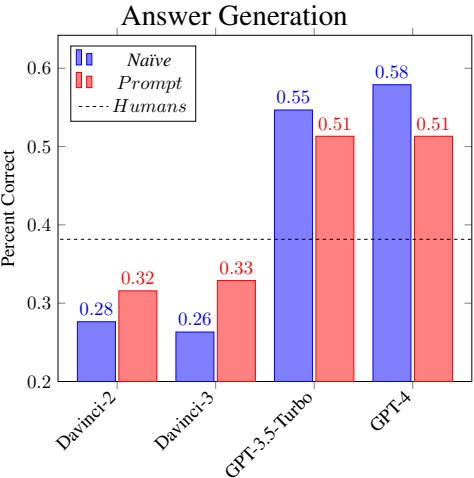

Figure 1: performance of the different models. Human performance is presented in the dashed line.

correct answer from the ground-truth. The models did not show different performances [F(3,3)=2.43, p=0.24], nor did the models' previous success in solving the stumper [F(1, 3) = 5.25, p = 0.106].

For the Model's responses (Figure 3), a three-way ANOVA was conducted, with model type, the correctness of the response, and the prompt type. Here, too, no effects were found for the model type [F(3,10)=1.28 p>0.33] or the prompt [F(1,10)=0.13, p>0.7]. The model's previous success has significantly improved performance [F(1,10)=24.97, p=0.0005].

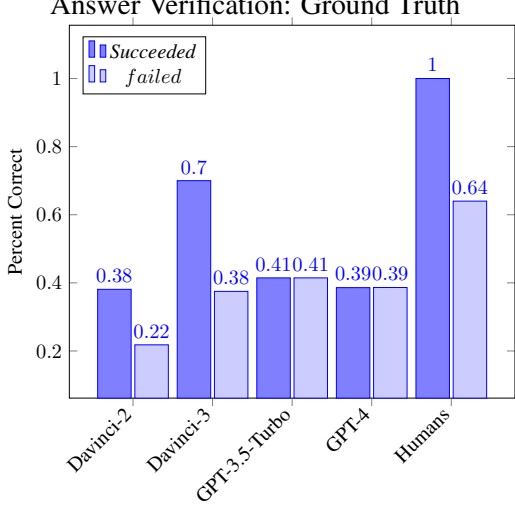

Figure 2: Accuracy of the models and humans in choosing the correct answer out of two alternatives.

## 5 Discussion and Conclusions

This study compared the stumper-solving abilities of LLMs and humans. We found that while the

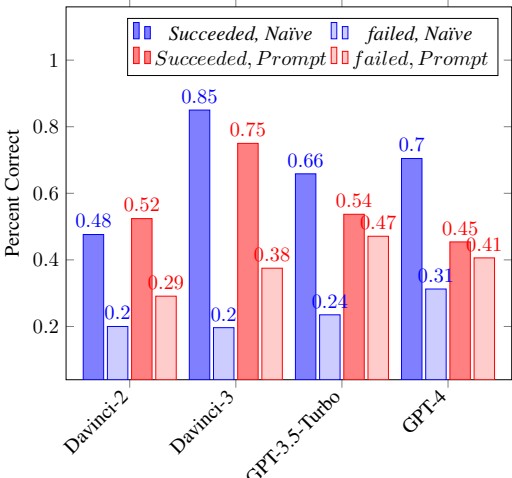

Figure 3: models' accuracy in choosing a solution between their previous response and the ground truth.

LLMs are better than humans at solving stumpers, their answer-verification abilities are limited and fall far from human performance. These findings provide valuable insights into the capabilities and limitations of LLMs, their relationship with human cognition, and the potential for utilizing LLMs as a framework for cognitive capacities.

The results showed that LLMs, specifically the LLMs used in this study, demonstrated proficiency in solving stumpers that are obstructed by misleading representations. The models correctly solved 26%-58% of the stumpers, outperforming human participants and surpassing the chance level (Figure 1). This suggests that LLMs possess the skills required for solving these types of questions.

The study revealed an improvement in performance for more advanced models. The chat models, GPT-3.5-Turbo and GPT-4, which are fine-tuned with human feedback during training, outperformed the GPT-3 models (Davinci-2 and Davinci-3). This indicates that advancements in model training contribute to enhanced problem-solving abilities. Prompting the models with additional pairs of riddles and their ground-truth answers had a positive impact on the performance of the GPT-3 models, further emphasizing the importance of context and prior knowledge in solving stumpers.

Despite their ability to generate correct answers, the models fell short in the task of answer verification compared to human participants (Figure 2). To further stress this problem, we have asked the models to compare their correct responses against a false response (Figure 3, full bars), demonstrating their inconsistency and inability to verify an-

swers. Humans, however, demonstrated a higher proficiency in recognizing the correct answer, even when they were unable to solve the problem initially (Figure 2). This suggests that humans possess a verification capability, considered a System 2 process, which has not yet fully emerged in LLMs. Interestingly, the Davinci-3 model showed good performance in recognizing correct answers curated by the authors (70%; Figure 2) and by itself (85%; Figure 3).

The overall pattern of results suggests that GPT-3 aligns better with human capabilities, as its answer-verification capabilities are better than its solving capabilities. This pattern stands in contrast to GPT-3.5 and GPT-4, which solve stumpers better than they verify their solutions. This finding indicates that for disciplines interested in using LLMs to model human behavior or cognition, Davinci-3 is likely a more suitable model to employ. This is in line with (Hagendorff and Fabi, 2023), which shows how GPT-3 (but not GPT-4 and GPT-3.5) exhibits similar biases and errors as demonstrated by humans. Another reason to consider Davinci-3 over GPT-4 and GPT-3.5 in modeling human behavior is the fact that the results acquired here, as well as the psychological literature, suggest that it is easier for humans to classify a correct response than generates it (Pintrich, 2002), a pattern of result similar to Davinci-3 and not congruent with GPT-4 and GPT-3.5 performance. This is closely related to the literature showing that recognition is considered easier than recall, as the former requires only identifying the presence of familiar information, whereas the latter demands retrieving specific information from memory without external cues (Roediger III and Karpicke, 2006).

The challenge of answer verification is closely related to the problem of text classification, which has been found to be challenging for LLMs (Sun et al., 2023). There is a significant discrepancy between the abilities to generate a correct answer and to classify a correct response. This has important implications for estimating LLM capabilities, as many NLP benchmarks are designed based on the model's ability to classify correct answers (Rajpurkar et al., 2016; Wang et al., 2018; Dagan et al., 2005; Reddy et al., 2019; Clark et al., 2019). One possible implication is the necessity of including interaction-based measures (Collins et al., 2023), based on continuous human-LLM interaction, when evaluating LLMs. Like in oral ex-

ams, the opportunity to react to the models' output in tailored follow-up questions allows the evaluator a deeper probing into the models' capabilities (Gharibyan, 2005; Davis and Karunathilake, 2005).

Furthermore, the findings from this study can inform the development of benchmark tasks for evaluating the intelligence and human-like behavior of LLMs. Stumpers provide a challenging domain that tests problem-solving, associative capacities, and verification skills. By designing more comprehensive benchmarks and evaluating LLMs' performance on these tasks, we can gain a better understanding of their cognitive capabilities and identify areas for improvement.

In conclusion, this study investigated the ability of large language models (LLMs) to solve stumpers, challenging riddles characterized by elusive solutions. Our findings demonstrate that LLMs, especially the advanced models GPT-3.5-Turbo and GPT-4, exhibit a remarkable proficiency in solving stumpers, surpassing human performance in some cases. These results highlight the potential of LLMs as powerful problem-solving tools and provide insights into the cognitive processes involved in solving complex puzzles. Our analysis also uncovered that the human ability to verify solutions has not been fully developed yet in LLMs. Future research can build upon these findings to explore the role of context, expand the variety of stumpers, and investigate the generalizability of LLMs in different domains, contributing to developing more robust and human-like artificial intelligence.

## 6 Limitations

Our study on stumpers and large language models (LLMs) has several limitations to consider. Firstly, the limited number of 76 validated stumper, even with two more versions of each in the enhanced dataset, potentially restricting the representativeness and generalizability of our findings. Secondly, we focused exclusively on OpenAI models, limiting the scope of comparison with other language models. Thirdly, subjective judgment was involved in evaluating correctness, leading to potential variations in interpretations. Lastly, prompt engineering techniques were underutilized, potentially limiting the models' problem-solving potential. Future research should address these limitations for more robust and comprehensive insights into LLMs' problem-solving abilities.

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

# Appendix

## A Stumpers examples

**1** A father and son were involved in a traffic accident. The father was killed, and the son was rushed to hospital.
The surgeon walked into the operating room, and upon seeing the severely wounded boy cried out: "OMG, it is my son!".
How could this be true?

Answer: *The surgeon is the boy's mother*

**2** A very tall man was holding up a wine decanter way above his head.
He let go of it, and it dropped to the carpet he was standing on.
Explain briefly how not a single drop of wine was spilled.

Answer: *The decanter was empty*

**3** Farmer Joe eats two fresh eggs from his own farm for breakfast every day.
Yet there are no chickens on his farm. Where does Farmer Joe get his eggs?

Answer: *Famer Joe does not eat chicken eggs, but a different animal's egg, such as ducks.*

**4** Marcy went from one bank of a river to the one 20 meters across.
There are no bridges on the river.
Marcy had no equipment, no devices, no special clothing, and she can't even swim.
She relied on her own body only -and none of it got wet!
Explain briefly how she managed this.

Answer: *The river was dry.*

## B Enhanced data-set results

Figure 4: Perfomrnace scores for the different models on the enhanced data-set. The dashed line indicates human performance in the original set.

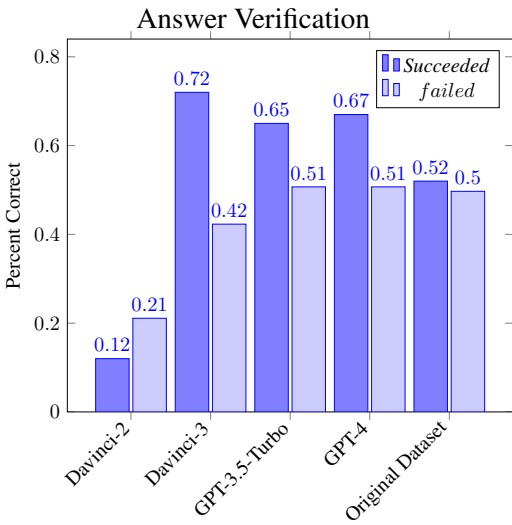

Figure 5: Accuracy of the models in choosing the correct answer out of two alternatives.

## C  Prompt examples

**Answer Generation, Naïve, GPT-3 models**

```
An answer to a riddle is correct
only if it is consistent with
all the riddle's clues, sensical,
specific, logical, and fitting
with the context of the riddle.
--

Riddle:
A father and son were involved in
a traffic accident.  The father
was killed, and the son was
rushed to hospital.  The surgeon
walked into the operating room,
and upon seeing the severely
wounded boy cried out:  "OMG, it
is my son!".  How could this be
true?

Answer:
```

**Answer Generation, Prompt, GPT-3 models**

```
An answer to a riddle is correct
only if it is consistent with
all the riddle's clues, sensical,
specific, logical, and fitting
with the context of the riddle.
--
Riddle:
Long after the screen of Kim's
smart phone had cracked.  It
was still functioning just fine.
Before he could replace it, the
phone accidentally fell into the
family's swimming pool.  It was
retrieved almost at once, but
- alas - the phone was dead.
Yet no water had penetrated
the cracked screen, so all the
critical components remained
completely dry.  Explain briefly
why the phone was dead.

Answer:
the pool was empty

--
Riddle:
Polly bought a beautiful parrot.
The seller guaranteed that the
bird repeats everything it hears.
However, try as Polly might to
teach it, her squawking parrot
never repeated a single word.
The seller did not lie.  Explain
briefly.

Answer:
The parrot was deaf

--
Riddle:
An accountant says:  "That
attorney is my brother", and that
is true - they really do have the
same parents.  Yet the attorney
denies having any brothers - and
that is also true!  How is that
possible?

Answer:
```

**Answer Generation, naïve, Chat models**

```
{'model': 'gpt-3.5-turbo',
'messages': [
'role': 'system',
'content': "An answer to a
riddle is correct only if it is
consistent with all the riddle's
clues, sensical, specific,
logical, and fitting with the
context of the riddle.",
'role': 'user',
'content': 'Riddle:\nTwo
Italians are sharing a pizza.
The older Italian is the brother
of the younger Italian. But
the younger Italian is not the
brother of the older Italian.
Explain briefly. ',
'role': 'assistant',
'content': 'Answer:\n'}],
'temperature': 0.0,
'frequency_penalty': 1.0,
'presence_penalty': 0.5,
'n': 1,
'max_tokens': 120}
```

**Answer Generation, prompt, Chat models**

```
{'model': 'gpt-4',
'messages': [
{'role': 'system',
'content': "An answer to a
riddle is correct only if it is
consistent with all the riddle's
clues, sensical, specific,
logical, and fitting with the
context of the riddle."},
{'role': 'user',
'content': 'Riddle:\nCindy
recycles everything: paper,
glass, metal, plastic, etc. She
also brings her still useable
stuff (books, housewares, etc.
) to the donation bins at
the recycle center. Recently,
she brought a bag full of
large (2 liter) bottles to
the recycling center. The
volunteer on duty could barely
lift it. Explain briefly.
\n\nAnswer:\nThe bottles were
full. They were donations to the
plant.\n\n\n--\nRiddle:\nFred
bought a used car from his
neighbor next door. The neighbor
claimed that the car got 35 miles
per gallon. Fred, an excellent
driver, could only get about
half that much, in spite of
driving on the same roads as
his neighbor. Explain in a few
words. \n\nAnswer:\nThe seller
was lying\n\n\n--\nRiddle:\nTwo
Russians were standing in line.
The taller one was the brother of
the shorter one, but the shorter
one was not the brother of the
taller one. Explain in a few
words how that is possible. '},
{'role': 'assistant', 'content':
'Answer:\n'}], 'temperature':
0.0, 'frequency_penalty': 1.0,
'presence_penalty': 0.5, 'n': 1,
'max_tokens': 120}
```

**Answer Verification, GPT-3 models**

```
  An answer to a riddle is
correct only if it is consistent
with all the riddle's clues, be
sensical, specific, logical, and
fitting with the context of the
riddle.

Riddle:
Farmer Joe eats two fresh eggs
from his own farm for breakfast
every day. Yet there are no
chickens on his farm. Where does
Farmer Joe get his eggs?

Answers:
1. Famer Joe do not eat chicken
eggs, but a different animal's
egg, such as ducks.
2. Farmer Joe gets his eggs from
the grocery store.

Which of these answers is
correct?
```

**Answer Verification, Chat models**

```
  {'model': 'gpt-3.5-turbo',
'messages': [
{'role': 'system',
'content': "An answer to a
riddle is correct only if it is
consistent with all the riddle's
clues, be sensical, specific,
logical, and fitting with the
context of the riddle."},
{'role': 'user',
'content': "Riddle:\nAlex
is Bobbie's blood relative,
and Bobbie is Charlie's blood
relative, but Alex is not a
blood relative of Charlie. How
come? \n\nAnswers:\n1. Alex
and Charlie could be Bobby's
parents, making them both Bobby's
blood relatives but not each
other's\n\n 2.Alex is Bobbie's
parent, and Bobbie is Charlie's
parent, but Alex is not a parent
of Charlie.\n\nWhich of these
answers is correct?"},
{'role': 'assistant',
'content': 'The correct answer
is number '}],
'temperature': 0.0,
'frequency_penalty': 1.0,
'presence_penalty': 0.5,
'n': 1,
'max_tokens': 20}
```
