# OpenReview forum: "Decoding Stumpers: Large Language Models vs. Human Problem-Solvers"
_EMNLP/2023/Conference — EMNLP 2023 Findings_

### Official Review · Reviewer_WvWv · 2023-07-29

**Soundness:** 2

**Excitement:**

3: Ambivalent: It has merits (e.g., it reports state-of-the-art results, the idea is nice), but there are key weaknesses (e.g., it describes incremental work), and it can significantly benefit from another round of revision. However, I won't object to accepting it if my co-reviewers champion it.

**Missing References:**

I did not look into the missing references because of the major concerns in this paper (see Reasons To Reject and Questions For The Authors)

**Paper Topic And Main Contributions:**

The paper attempts to experiment with a few LLMs on a dataset of cognitive stumpers and evaluate the performance between LLMs and humans.
The experiments consist of two parts: answer generation and answer verification.
The authors claim that LLMs' task-solving ability outperforms humans while humans exhibit better verification capabilities.

**Questions For The Authors:**

A. What is the evaluation metric for the experiments? The authors mentioned 'two authors evaluated the responses unanimously' (line 138), but how? and based on what metric? The stumpers seem to be open-ended and can have many different possible answers.

B. No details of the dataset. It is possible (some/all) the models have already seen these stumper questions in their training corpus, and the models remember the answers instead of performing 'thinking'. Can authors provide justifications?

C. In section 3.2, it seems the authors just changed the original open-ended QA task into a binary-choice task. Why the change of task format becomes answer verification?

D. In section 3.2, the authors mentioned 'humans were presented with the two possible solutions'. Where do these alternative solutions come from? These alternatives may be easy to distinguish and are not suitable for evaluation.

E. In section 3.3, why the design of the answer verification task for the models is different from the one for humans? Please provide justifications.

F. In Figure 2, Davinci-3's success rate is abnormally high compared with the other models and its answer generation performance in Figure 1. Did the authors further analyze the reasons? If so, please provide justifications. (This question is more related to the concern of novelties, it seems the authors just 'presented' the results without actually analyzing the experiment scientifically)

G. For asking each human subject to answer one single question in the dataset, the authors mentioned 'to avoid learning' in line 114. It seems unreasonable. Are the authors suggesting humans can learn how to answer new stumpers by seeing any other stumpers? If so, this hurts the claim that the task need 'creative thinking' and 'hard for humans to solve' (line 75). Please provide justifications for this design.

H. The results in Figures 2&3 show Davinci-3 outperforms GPT-3.5 and GPT-4 in the binary-choice task, but the authors claim GPT-3.5 and GPT-4 'outperformed the GPT-3 models' (line 204). This is contradictory. Please provide justifications.

I. The authors claimed that few-shot prompting improves GPT-3's performance, thus justifying 'the importance of context and prior knowledge in solving stumpers'. However, few-shot prompting leads to a performance drop on GTP-3.5 and GPT-4, which is contradictory. Please provide justifications.

**Reasons To Accept:**

The analysis of LLM's ability for solving stumpers seems to be an interesting research direction, but I would emphasize my concerns in Reasons To Reject.

**Reasons To Reject:**

I have three primary comments regarding this paper (ranked by importance)

(1). The novelty in this paper does not meet a top-venue standard.
- Essentially, the authors took an existing dataset, used OpenAI's API to run some evaluations, recruited humans to do the same task, and plotted some diagrams. Instinctively, the contribution in the paper seems to be just a section of a regular top-venue paper or a benchmarking section for the dataset paper.
- The experiments are too simple and not well-designed (see (2)), leading to (3), where the paper is filled with less-relevant content and discussions but with minor novel discoveries and contributions.

(2). The experiments are not well-designed and organized.
- No details of the design of the prompts. As NLP experts, the authors need to justify the design of the prompts to make the best use of LLMs for solving the stumpers.
- No details of the dataset (the authors only mentioned the dataset size in the limitation section!). The quality of the dataset is questionable, especially for the ground-truth annotations. By looking at the samples in Appendix A and B, it seems each stumper question can have a lot of different reasonable answers, but the dataset only has one answer for each data. (see question B/D below). In addition, missing a lot of details for the experiment design (see question A/E below)
- The human study design is questionable. The human performance should not be evaluated by asking each human subject to answer one question in the dataset, which can cause reliability issues with the evaluation result. (see question G below)
- The answer verification section is not a verification task. Instead, the authors just converted the original QA task into a binary-choice task. (see question C below) For a verification task, I expect to see the authors ask models and humans to provide reasoning to verify whether their provided answers are correct or not.

(3). The paper structure is not well-organized.
- Authors spend a whole page in a short paper to describe the difference between 'stumpers' and system1&2 (which are not directly related to the experiment in this paper) and another full page to provide discussions, while only using half a page to describe the experiment details and another half page to report the experiment discoveries (exclude the spaces for the figures). Too much text to describe less-relevant content and hypothesis makes the whole paper read like a blog post instead of a scientific paper, leading to a lack of novelty.
- The details of the dataset are missing (By only reading the paper, I have no idea of this dataset's quality, validity, and stats), leading to concerns about the dataset quality. E.g., how the ground-truth is collected? Can it faithfully reflect the performance of LLMs and humans?
- Many experiment design decisions are missing supporting evidence (see questions below), which leads to a potentially over-claiming issue (see (1)).

Because of the rebuttal word limit, I don't expect the authors to be able to address all my questions and concerns, but I'd like to use the questions as suggestions for the authors to improve their work.

**Reproducibility:**

3: Could reproduce the results with some difficulty. The settings of parameters are underspecified or subjectively determined; the training/evaluation data are not widely available.

**Reviewer Confidence:**

4: Quite sure. I tried to check the important points carefully. It's unlikely, though conceivable, that I missed something that should affect my ratings.

**Typos Grammar Style And Presentation Improvements:**

I did not look into the typos because of the major concerns in this paper (see Reasons To Reject and Questions For The Authors).

I would like to emphasize that the writing of this paper is not well-organized. For a short paper, I expect to see all the key justifications, experiment design decisions, results, and important outcomes in the paper, not 2 pages of intro and discussions. If the authors are spending a lot of content for discussions, the authors should provide supporting evidence from the experiments to support the 'hypothesis' in discussions.

---

> ### Author Rebuttal · Authors · 2023-08-29
>
> Dear Reviewer WvWv,
>
> We sincerely appreciate your thorough review of our paper and your insightful feedback. Your detailed observations and questions provide us with valuable opportunities to improve the clarity, rigor, and depth of our research. We are committed to addressing each of your concerns to enhance the quality and contribution of our work.
>
> Reasons to Reject:
>
> (1). The novelty in this paper does not meet a top-venue standard.
>
> We understand your perspective regarding the novelty of our paper. While the core of our work lies in the evaluation of Large Language Models (LLMs) on stumpers, we recognize that the contribution might appear limited. However, we believe the unique nature of stumpers and the insights we draw from LLMs' performance offer valuable implications for understanding cognitive processes and the capabilities of LLMs. We will reemphasize these points to ensure the significance of our work is better articulated.
>
> (2). The experiments are not well-designed and organized.
>
> We apologize for the lack of details in our paper regarding prompt design and dataset. Prompt design is indeed crucial for obtaining reliable results from LLMs. We will provide a more comprehensive explanation of the prompt design and justification for our choice in the final version, given the opportunity. The prompts used in this study were the same instructions given to the human participants. Other, less intuitive prompts, were tested after the original one, and yielded similar results (including the use of chain-of-thought). Due to the lack of space in this short paper, and since prompt engineering is not the focus of this project, we did not provide further details about how we built the prompts, and we will amend it in the final version.
>
> As for the dataset, we acknowledge that more information is required to assess its quality and validity. In the extra page of the final version, we will include details about how the dataset was created and the process of collecting ground-truth annotations. Furthermore, we will make the entire dataset, including scoring, publicly available to address transparency concerns.
> We also appreciate your concern about the human study design. The constraint of asking each participant to solve only one question was imposed in order to mitigate the potential bias from shared mechanisms among stumpers. Moreover, past studies in the psychological literature have always limited each participant to one stumper for this reason, and we naturally wanted to be consistent with the literature. We acknowledge the potential reliability issues you've raised and will consider this aspect more carefully in our future research.
>
> As for the answer verification task, we apologize for the misuse of terminology. You are correct that a verification task should involve reasoning to verify the correctness of provided answers. We will correct this and ensure that our terminology aligns with the intended task. We are currently considering using “recognition” or “comparison”, and would greatly appreciate your feedback on it.
>
> (3). The paper structure is not well-organized.
>
> Your feedback on the paper's structure is duly noted. We agree that the current organization may lead to confusion and hinder the paper's readability. We will streamline the structure, ensuring that relevant content is allocated the appropriate space while maintaining a cohesive narrative that highlights the novelty of our findings. Specifically, we will add our reasoning of using the dual-systems model, as prevalent in social-sciences when modeling thinking process,  in the introduction while keeping it as short as possible, limit our discussion, and extend the details provided for the experiment details and discoveries.
>
> Questions for the Authors:
>
> A. What is the evaluation metric for the experiments?
>
> The evaluation metric for our experiments involves comparing the responses generated by LLMs and human participants to the ground-truth answers and computing the fraction of correct responses. In the case of answer generation, a "correct response" is defined as any answer that adheres to the given criteria for correctness stated in the paper (lines 109-112 “An answer to a riddle is correct only if it is consistent with all the riddle’s clues, sensical, specific, logical, and fitting with the context of the riddle”). In the answer verification task (that will be renamed), participants and models are presented with pairs of possible answers, one correct and one incorrect. The performance metric is based on their ability to choose the correct answer from these options.
>
> B. No details of the dataset.
>
> We apologize for the lack of details about the dataset. As previously mentioned, the dataset is an exhaustive list of stumpers curated by Bar-Hillel (2021) with possible correct and incorrect responses provided by the authors. We will provide a comprehensive explanation of the dataset's creation process and the methods used to ensure its quality and validity.
>
> As to the possibility that the LLM was exposed to the dataset during training, we addressed it in two ways. First, we asked the LLM to generate each of the stumpers in the dataset given their first sentence and observed poor results. Then, we also noticed, as reported in the paper, that LLMs do not perform well on recognizing the correct answer to a stumper when asked to choose between two answers. This is an indication that even if the model was exposed to the dataset during training, it was not able to memorize it properly. Finally, we note that the stumpers dataset was published in 2021 so it is possible that it was not included in the GPT training set. We thank the reviewer for this important comment and given the opportunity we will address it in the extra page allowed in the final version of the paper.
>
> C. In section 3.2, why the change of task format becomes answer verification?
>
> Again, your comment about the inaccurate naming of this process was precise, and we will amend it. The task change from answering (or generating a response to) the stumper to judging the answers’ correctness, was introduced to highlight the model’s lack of “understanding”. Stumpers, by their definition, become very trivial once the solution is given (i.e. it is easy to recognize the correct answer once it is presented). Judging the truthfulness of a solution is a very simple task humans can do almost at ceiling level, yet the models could not. When asking the models to rule “correct” or “incorrect” on answers, each model had their bias (either to mostly agree or disagree). For this reason, we used the comparison task instead of asking for agreement.
>
> D. In section 3.2, the authors mentioned 'humans were presented with the two possible solutions'.
>
> The alternative solutions were crafted by the authors to be (a) semantically relevant to the stumpers while (b) being incorrect according to the criteria for a correct response. Both the correct and the incorrect answers were presented to both human participants and LLMs to ensure a fair comparison. We will add to the discussion the need to systematically test different distractors in this task in order to ensure the results on different effort levels of the task.
>
> E. In section 3.3, why the design of the answer verification task for the models is different from the one for humans?
>
> The unequal design of the answer verification task was motivated by the difference in the capabilities of humans and models. While models can generate responses independently, human participants remember recent experiences, which make an exact design impossible. This design choice was made in order to address this difference and ensure a more accurate comparison.
>
> F. In Figure 2, Davinci-3's success rate is abnormally high compared to the other models.
>
> We appreciate your observation. While we did notice this anomaly, which was persistent across prompts, further analysis of this particular pattern falls out of the scope of this short paper. It's an interesting avenue for future research to delve into the reasons behind this disparity in performance among the models.
>
> G. For asking each human subject to answer one single question in the dataset...
>
> Your concern regarding the human study design is valid. Some stumpers share common mechanisms, which can introduce a potential bias when participants are exposed to multiple similar problems. Like many other psychological tests (e.g., the bat-and-the-ball, optical illusions), seeing the solution for one may bias (in favor or against) finding the solutions in the next. This tendency does not mean that the tasks are easy or that they do not call for creativity, it just points out that some cues can be (too) informative and should be avoided for the fairness of the comparison. Therefore, and following established psychological practices, we opted for a design where each participant encountered a single stumper to avoid potential learning effects.
>
> H. The results in Figures 2&3 show Davinci-3 outperforms GPT-3.5 and GPT-4 in the binary-choice task...
>
> We apologize for any confusion. Your observation is correct, and we will clarify the text to accurately reflect the model performances based on the specific tasks.
>
> I. The authors claimed that few-shot prompting improves GPT-3's performance...
>
> We appreciate your insight into this matter. The performance variation introduced by few-shot prompting across different models indeed requires a more thorough reporting. We will acknowledge this behavior more explicitly in the revised version.
>
> Once again, we sincerely thank you for your thoughtful feedback and questions. Your insights have significantly enriched our understanding of potential improvements, and we are fully committed to addressing these concerns in the revised version of our paper.
>
> Best regards,
> Submission668 authors

---

### Official Review · Reviewer_zMZm · 2023-08-05

**Soundness:** 3

**Excitement:**

3: Ambivalent: It has merits (e.g., it reports state-of-the-art results, the idea is nice), but there are key weaknesses (e.g., it describes incremental work), and it can significantly benefit from another round of revision. However, I won't object to accepting it if my co-reviewers champion it.

**Missing References:**

I think you should cite Daniel Kahneman's book "Thinking, Fast and Slow" or equivalent paper where two systems come from.

**Paper Topic And Main Contributions:**

This paper proposes a new task, solving stumpers. Here, the authors say that stumpers are sort of riddle that has misleading cue which sparks a certain visual or semantic representation. This paper proposed a new dataset with 76 problems and evaluated performance of LLMs on solving such stumpers. As a result, this paper suggested that LLMs exhibits better ability of solving stumpers but worse ability of identifying right answers for the stumpers compared to human.

Main contributions:
- Proposing a new task, solving stumpers and provide a small dataset to test LLMs
- Demonstrate two ways to analyze LLM's performance on stumpers: Answer Generation and Answer Verification.

**Questions For The Authors:**

Thanks for your hard work. To make it clear what you're saying in this paper, please answer these three questions.

Question A. I think the paper does not clearly states what a stumper is. Though I understood it as a riddle that has some tricky things that makes people fail to answer it, but it seems yet vague to me. Could you give me an operational or formal definition of it?

Question B. Why do you use the notion of System 1 and 2 (I assume that you're inspired by Daniel Kahneman)? The relationship between these two systems and your stumper task is not clearly stated in the paper.

These are questions about some logical gaps in the discussion.

Question C. To clearly state one condition is better than the other after ANOVA, we need to run post-hoc tests. The ANOVA just says that the distribution of two conditions are differ (not the direction of which thing is larger). Could you report the corresponding results?

Question D. I think two statements lines 071-073 and lines 222-224 are in conflict. Here, you said that stumpers both do not fall within two systems but related to system 2. Please correct my view if I'm wrong.

Question E. I think the reported result are irrelevant to lines 266-273. I cannot see any relationship between "interaction-based measures" and generation / verification procedure. Could you explain further?

**Reasons To Accept:**

- Propose a new way to analyze LLM's linguistic or common-sense-related ability.
- Demonstration of such ways and identifying some limitations of current LLMs

**Reasons To Reject:**

- The paper is not well written: the authors should re-organize the paper to clearly deliver what they did. There are several equivocations about stumpers or riddles, which prohibits understanding of this paper.
- The datasize of 76 is somewhat small, though the dataset is designed for testing purposes and I know that the authors already stated this point. To reflect diverse linguistic variations or situations, I think the task requires at least 200 stumpers.
- The discussion has some logical gaps. Please refer to the questions sections for the logical gaps.

**Reproducibility:**

4: Could mostly reproduce the results, but there may be some variation because of sample variance or minor variations in their interpretation of the protocol or method.

**Reviewer Confidence:**

3: Pretty sure, but there's a chance I missed something. Although I have a good feel for this area in general, I did not carefully check the paper's details, e.g., the math, experimental design, or novelty.

**Typos Grammar Style And Presentation Improvements:**

- Some of the required things to understand this paper stated in the Appendix. Please state some example classes of stumpers in Section 2 and (if possible) provide a short-version of prompt in Section 3.
- Distinction between Section 3.2 and 3.3 makes me confused. Isn't 3.3 are subset of 3.2?
- There are several equivocations and repetitions in the paper, such as 'stumper'. Please adjust them.

---

> ### Author Rebuttal · Authors · 2023-08-29
>
> Dear Reviewer zMZm,
>
> We sincerely appreciate your thoughtful and detailed review of our paper. Your feedback is invaluable in enhancing the quality and clarity of our work. We have carefully considered each of your comments and suggestions, and we would like to address them individually.
>
> First, we would like to acknowledge your concern about the paper's organization. We apologize for any confusion caused by the current structure and assure you that we will reorganize the paper to provide a clearer narrative flow. Your feedback underscores the importance of delivering our methodology, findings, and discussions in a well-structured manner that facilitates a cohesive understanding of our research.
>
> As for the dataset size, we would like to clarify that the 76 stumpers used in our study represent an exhaustive list taken from the Bar-Hillel 2021 paper, the definitive paper about stumpers. While we acknowledge the limitation of the dataset size, the generation of new stumpers is a complex and non-trivial task, a challenge that many psychologists have grappled with. Also, such newly created stumpers would, by definition, have a different status than the original 76. We hope that if the NLP community becomes interested in this type of stimuli it  will encourage a multidisciplinary effort to curate a bigger dataset.
>
> Question A: We appreciate the reviewer's request for a clearer explanation of stumpers and their selection in our study. “A stumper is a riddle, the solution to which is typically so elusive that it does not come to mind, at least initially - leaving the responder stumped… [stumpers] work by eliciting a representation of the situation described in the narrative, which then blocks the solution ” (Bar Hillel, 2021). In other words, stumpers are a specific type of riddle that poses two key characteristics. Firstly, they present a notable challenge, often leaving individuals stuck and unable to find any solution, even if incorrect. Secondly, their solutions are remarkably straightforward - once revealed, they are immediately recognized by the solver.  These unique attributes make stumpers an intriguing subject of study as they reveal the capacity of humans and language models to not merely interpret language but to reinterpret it in a way that resolves a problem. We will certainly provide a more explicit explanation of this definition in the revised paper.
>
> Question B: Your observation about the relationship between System 1 and System 2 and the stumper task is accurate. We recognize that we didn't adequately establish the connection between the systems and our hypotheses within the paper. The dual-system model of the mind has been the most prevalent model of thought and behavior in psychology, economics and social science in general (Goldstein & Young, 2022; Sherman, Gawronski & Trope, 2014), especially in addressing systematic limitations of cognitive and artificial systems (Kahneman, 2013).
> Our main point in mentioning the dual systems model is that the stumper task represents a distinctive challenge that doesn't fit neatly into the System 1 and 2 dichotomy. Particularly, stumpers are a good example of a failure of both systems, with System 1 failing to provide a swift response, even if incorrect, and System 2 lacking a clear procedure for a solution. To tackle stumpers, a third process emerges, involving reinterpreting the problem's description to enable a solution. This unique contribution of the stumper dataset showcases the limitations of the existing framework, which is why we chose to focus on it. We will address this motivation more comprehensively in the final version, given the opportunity.
>
> Question C: We apologize for not including post-hoc test results after ANOVA. We will ensure that the corresponding post-hoc test outcomes are reported in the final version of the paper.
>
> Question D: Your observation regarding the apparent conflict between lines 071-073 and lines 222-224 is astute. In lines 071-073, we highlight that stumpers do not neatly fall within either System 1 or System 2, as we further elaborate in response to question B. On the other hand, lines 222-224 pertain to the recognition of potential solutions by humans and not to generating them. This distinction implies that while generating responses may not align with System 1 or 2, recognizing solutions could be associated with System 2-like processes. We will clarify this distinction in the revised manuscript. Thanks very much for highlighting this.
>
> Question E: In the paper, we demonstrate that models can provide suitable answers but often struggle in making accurate judgments, while humans excel at recognizing solutions. Lines 266-273 hint at the potential for interaction-based measures to improve this situation.
>
> We greatly appreciate your suggestions regarding missing references, typos, grammar, style, and presentation improvements. We will certainly incorporate your recommendations to enhance the clarity and coherence of our paper. Moreover, we will address the issues you highlighted in the Appendix and strive to eliminate any equivocations or repetitions.
>
> Once again, thank you for your diligent review. Your insights have significantly contributed to refining our work, and we are committed to addressing all the points you've raised to enhance the quality and rigor of our paper.
>
> Best regards,
> Submission668 Authors

---

### Official Review · Reviewer_2ch8 · 2023-08-12

**Soundness:** 3

**Ethical Concerns:**

Yes

**Excitement:**

4: Strong: This paper deepens the understanding of some phenomenon or lowers the barriers to an existing research direction.

**Justification For Ethical Concerns:**

IRB information is required.

**Paper Topic And Main Contributions:**

This short paper entitled “Decoding Stumpers: Large Language Models vs. Human Problem Solvers” has a potential to improve the ability of LLMs and can contribute to NLP engineering as well as new dataset addition. Identifying a stumper itself is already a contribution. Additionally, the authors found out that the current LLMs are excellent solvers for stumpers in comparison with human solvers.

**Questions For The Authors:**

A. How would you define a stumper problem? How did you choose 76 stumper problems? What procedure did you use to choose the stumper problems?
B. Unlike other problems from professional exams such as LSAT bar test and SAT, stumper problems are hard to be standardized. Is there any strategy that you can standardize the stumper problems? Also, is there any way to make sure the abilities of the human solvers are similar?
C. In your study, human subjects are involved. Please, provide IRB information.


**Reasons To Accept:**

This short paper contributes to the NLP community by adding new stumper datasets and by comparing the performances of human solver and LLMs empirically.

**Reasons To Reject:**

However, this short paper may have too small samples to generalize the results. Additionally, unlike LLMs, human solvers are hard to be similar. There should be a way to control human solvers’ abilities.

**Reproducibility:**

5: Could easily reproduce the results.

**Reviewer Confidence:**

4: Quite sure. I tried to check the important points carefully. It's unlikely, though conceivable, that I missed something that should affect my ratings.

---

> ### Author Rebuttal · Authors · 2023-08-29
>
> We sincerely appreciate your thoughtful evaluation of our paper and your valuable comments. Your insights have provided us with an opportunity to address critical concerns and enhance the robustness and credibility of our research. We are committed to addressing each of your points and ensuring that the revised paper is more comprehensive and well-grounded.
>
> Reasons to Reject:
> Your concerns about the sample size and the potential difficulty in generalizing our results are duly noted. We would like to point out that we have used an exhaustive list of all available stumpers (found in Bar Hillel 2021).
> The problem with human’s lack of consistency is a general issue  in the psychological literature. In all behavioral experiments we treat two sources of variances (or inconsistencies), individual differences (the real difference between different participants) and noise (the fact that the participants are not necessarily consistent with themselves). In order to establish the reliability of our results we compared the human performance we acquired with the results previously attained. We mention it in the text stating: “All models correctly solved more than 25% of the stumpers, compared to human accuracy of 35% in (Bar-Hillel, 2021),  and 38.15% in our sample.” [142-144]. This shows in the group level the human performance is relatively stable.  We will note that our results replicate previous research and fall within the lines of human behavior experimental design.
>
> Questions for the Authors:
> A. How would you define a stumper problem? How did you choose 76 stumper problems? What procedure did you use to choose the stumper problems?
>
> We appreciate the reviewer's request for a clearer explanation of stumpers and their selection in our study. “A stumper is a riddle, the solution to which is typically so elusive that it does not come to mind, at least initially - leaving the responder stumped… [stumpers] work by eliciting a representation of the situation described in the narrative, which then blocks the solution ” (Bar Hillel, 2021). In other words, stumpers are a specific type of riddle that poses two key characteristics. Firstly, they present a notable challenge, often leaving individuals stuck and unable to find any solution, even if incorrect. Secondly, their solutions are remarkably straightforward - once revealed, they are immediately recognized by the solver. These unique attributes make stumpers an intriguing subject of study as they reveal the capacity of humans and language models to not merely interpret language but to reinterpret it in a way that resolves a problem. We will certainly provide the above mentioned definition in the revised paper. The 76 stumper problems used in our study were a comprehensive list taken from Bar-Hillel’s (2021) paper listing all researched verbal stumpers.
>
> B. Unlike other problems from professional exams such as LSAT bar test and SAT, stumper problems are hard to be standardized. Is there any strategy that you can standardize the stumper problems? Also, is there any way to make sure the abilities of the human solvers are similar?
>
> Standardizing stumper problems indeed presents challenges due to their propensity to induce cognitive blocks. To achieve a level of standardization, we provided a well-defined criterion for what constitutes a correct response to a stumper (lines 109-112: “An answer to a riddle is correct only if it is consistent with all the riddle’s clues, sensical, specific, logical, and fitting with the context of the riddle”). This criterion was communicated to both the models and to the human participants and the judges of the answers, ensuring a common basis for evaluation. Concerning the abilities of human solvers, while it is challenging to ensure complete similarity, we judged each response by the same definition, allowing multiple different responses to be correct. The stumpers were scored using 2 independent judges with above 95% agreement, ensuring the scoring is as standardized as human behavior can be.
>
> C. In your study, human subjects are involved. Please, provide IRB information.
>
> We highly value the ethical aspects of our study involving human participants. Our research was conducted with strict adherence to ethical guidelines, and since no personal-identifiable information was collected, nor any harm was done to participants it was eligible for Institutional Review Board (IRB) exemption. Throughout the study, we upheld the principles of participant privacy, informed consent, and ethical integrity. Given the opportunity, we will address this issue in the extra page allowed in the final version.
>
> Once again, we genuinely appreciate your feedback, which has been instrumental in identifying areas for improvement. Your insights will contribute significantly to the enhancement of our paper, and we are dedicated to incorporating your suggestions to produce a more comprehensive and credible final version.
>
>
> Best regards,
> Submission668 authors.

---

### Meta-Review · Area_Chair_X5hA · 2023-09-28

**Recommendation:** 2

**Metareview:**

The paper investigates the problem-solving capabilities of Large Language Models (LLMs) by evaluating their performance on stumpers, unique problems that pose challenges for human solvers but are easily verifiable. The paper compares the performance of four state-of-the-art LLMs (Davinci-2, Davinci-3, GPT-3.5-Turbo, GPT-4) to human participants. The paper finds that the new-generation LLMs excel in solving stumpers and surpass human performance. However, humans exhibit superior skills in verifying solutions to the same problems. The paper enhances our understanding of LLMs’ cognitive abilities and provides insights for enhancing their problem-solving potential across various domains.

Pros:
The paper addresses an interesting and novel research question: how well can LLMs solve stumpers compared to humans?
The paper uses a large and diverse set of stumpers from various sources and domains, covering logic, math, language, trivia, and creativity.
The paper employs rigorous experimental methods and statistical analyses to compare the performance of LLMs and humans on both solving and verifying stumpers.
The paper discusses the implications of the findings for the design and evaluation of LLMs, as well as the ethical and social issues raised by their problem-solving abilities.

Cons:
The paper does not provide a clear definition or formalization of what constitutes a stumper, or how to measure its difficulty or uniqueness.

---

### Decision · Program_Chairs · 2023-10-07

**Decision:**

Accept-Findings

**Comment:**

The paper investigates the problem-solving capabilities of Large Language Models (LLMs) by evaluating their performance on stumpers, unique problems that pose challenges for human solvers but are easily verifiable. The paper compares the performance of four state-of-the-art LLMs (Davinci-2, Davinci-3, GPT-3.5-Turbo, GPT-4) to human participants. The paper finds that the new-generation LLMs excel in solving stumpers and surpass human performance. However, humans exhibit superior skills in verifying solutions to the same problems. The paper enhances our understanding of LLMs’ cognitive abilities and provides insights for enhancing their problem-solving potential across various domains.

Pros:
The paper addresses an interesting and novel research question: how well can LLMs solve stumpers compared to humans?
The paper uses a large and diverse set of stumpers from various sources and domains, covering logic, math, language, trivia, and creativity.
The paper employs rigorous experimental methods and statistical analyses to compare the performance of LLMs and humans on both solving and verifying stumpers.
The paper discusses the implications of the findings for the design and evaluation of LLMs, as well as the ethical and social issues raised by their problem-solving abilities.

Cons:
The paper does not provide a clear definition or formalization of what constitutes a stumper, or how to measure its difficulty or uniqueness.